# Modeling the Genetic Basis of Individual Differences in Susceptibility to Gulf War Illness

**DOI:** 10.3390/brainsci10030143

**Published:** 2020-03-02

**Authors:** Byron C. Jones, Diane B. Miller, Lu Lu, Wenyuan Zhao, David G. Ashbrook, Fuyi Xu, Megan K. Mulligan, Robert W. Williams, Daming Zhuang, Carolina Torres-Rojas, James P. O’Callaghan

**Affiliations:** 1Department of Genetics, Genomics and Informatics, Department of Pharmacology, University of Tennessee Health Science Center, 71 South Manassas Street, Memphis, TN 38163, USA; llu@uthsc.edu (L.L.); wenyuanlabuse@uthsc.edu (W.Z.); dashbroo@uthsc.edu (D.G.A.); fxu10@uthsc.edu (F.X.); mmulliga@uthsc.edu (M.K.M.); labwilliams@gmail.com (R.W.W.); Dzhuang1@uthsc.edu (D.Z.); Ctorres9@uthsc.edu (C.T.-R.); 2Molecular Neurotoxicology Laboratory, Centers for Disease Control and Prevention, National Institute for Occupational Safety and Health, Morgantown, WV 26505, USA; jdo@cdc.gov

**Keywords:** BXD mice, recombinant inbred strains, candidate gene, DFP, neuroinflammation, corticosterone

## Abstract

Between 25% and 30% of the nearly one million military personnel who participated in the 1991 Persian Gulf War became ill with chronic symptoms ranging from gastrointestinal to nervous system dysfunction. This disorder is now referred to as Gulf War Illness (GWI) and the underlying pathophysiology has been linked to exposure-based neuroinflammation caused by organophosphorous (OP) compounds coupled with high circulating glucocorticoids. In a mouse model of GWI we developed, corticosterone was shown to act synergistically with an OP (diisopropylflurophosphate) to dramatically increase proinflammatory cytokine gene expression in the brain. Because not all Gulf War participants became sick, the question arises as to whether differential genetic constitution might underlie individual differences in susceptibility. To address this question of genetic liability, we tested the impact of OP and glucocorticoid exposure in a genetic reference population of 30 inbred mouse strains. We also studied both sexes. The results showed wide differences among strains and overall that females were less sensitive to the combined treatment than males. Furthermore, we identified one OP-glucocorticoid locus and nominated a candidate gene—*Spon1*—that may underlie the marked differences in response.

## 1. Introduction

In 1991 the USA sent about 700,000 military personnel, joined by another 200,000 from allied nations, to the Persian Gulf to counter the invasion of Kuwait by Iraq. Of those who participated in the conflict, 25–30% developed a multi-symptom malaise, Gulf War illness GWI [1,2]. Symptoms range from gastrointestinal complaints, to lethargy, cognitive lapses, and depression. As described by Dantzer and colleagues [3], sickness behaviors are likely linked to activation of macrophages and microglia, and subsequent neuroinflammation via production of proinflammatory cytokines such as *Tnfa*, IL1β, and IL6. GWI thus has the features of a neuroimmune disorder [4,5,6,7]; but causes are otherwise poorly understood. The leading suspected causes are chemicals to which the personnel were exposed. These include depleted uranium, cholinesterase inhibitors used as insecticides or as prophylactics against nerve agents (e.g., chlorpyrifos and pyridostigmine bromide) and even small amounts of sarin. The evidence points to exposure to the irreversible cholinesterase inhibitors chlorpyrifos and the nerve gas, sarin [8]. The former is an insecticide applied to living quarters and the latter was inadvertently released from ammunition dumps during their destruction by allied troops. GWI also presents two puzzles. The first is why only 25–30% became sick while all else being equal, the rest did not. This leads to the hypothesis that differential susceptibility to developing Gulf War illness is a gene-environment problem. To date, there has been only limited effort to address this problem. Georgeopolis and colleagues [6] proposed brain synchronicity indices as biomarkers of GWI and addressed individual differences in susceptibility as related to various HLA alleles. Alternatively, Steele and colleagues [9] differentiated individuals who had high butyrylcholinesterase activity from those with low activity. They demonstrated that those with the low activity and treated with pyridostigmine (a prophylaxis against organophosphate toxicants such as sarin) are at greater risk for developing GWI in response to wartime exposure than those evincing high activity. Nevertheless, at this time, there have been no published studies using GWAS or other genetic methods to delineate differential susceptibility to GWI. The second is why the illness has persisted for so long—nearly 30 years. The aim of this work is to address the former, using a genetic reference population of mice. An animal model of GWI was developed by O’Callaghan and colleagues [10]. The model involved exposing C57BL/6J (B6) mice to an irreversible cholinesterase inhibitor, diisopropylflurophosphate (DFP, a sarin surrogate). They proposed that exposure to DFP was necessary but not sufficient to produce neuroinflammation. Rather, high concentrations of circulating glucocorticoids (cortisol) as would be expected in a combat zone [11] were required as well. They showed that expression of proinflammatory cytokine genes was greatly enhanced by the co-exposure to DFP and corticosterone, the major glucocorticoid in rodents, compared to DFP alone [10]. Using this “two hit” treatment model, we will begin to address why all exposed troops did not become sick by expanding the testing of mice from the B6 strain to the genetic reference population of recombinant inbred (RI) strains derived from B6 and DBA/2J (D2) mouse strains (BXD) and both sexes. There are currently 150 extant BXD RI strains [12,13]. All have been genotyped and sequenced. The BXD family have an extensive phenome of over 7000 traits, including phenotypes for behaviors, immune parameters, neurochemistry, and pharmacology of alcohol and other drugs of abuse. Gene expression data have been recorded for numerous tissues including several brain regions. Moreover, there are ~6 million genetic variants (SNPs, insertions, deletions, duplications, etc.) that segregate in the family [13]. The strains are appropriate for systems genetics/systems biology analysis [14], genetic mapping and genetic correlations of parameter means, and thus constitute an ideal platform for toxicogenomic research [15]. All data are available at www.genenetwork.org. GeneNetwork exists in two forms, GN1 and GN2 [16]. GN2 is an expansion and refinement of the features of GN1. A tutorial of how to use GN1 may be found here: http://gn1.genenetwork.org/tutorial/ppt/index.html. The tutorials here can provide the basis for working in Gn2. Here, we report variability in cytokine gene expression following exposure to corticosterone (CORT) and DFP in 30 BXD recombinant inbred mouse strains and in both sexes.

## 2. Materials and Methods

### 2.1. Animals

The subjects for this study were male and female mice from 30 BXD.

Recombinant inbred strains. The animals were between 2 and 4 months of age at testing and 5–8 animals per strain, sex and treatment group were used. All animals were obtained from the breeding colony at the UTHSC vivarium. All animals had access to water and food ad libitum and controlled climate at 20 ± 2 °C and 35% relative humidity. All procedures were approved by the UTHSC animal care and use committee, approval code 17-022.0B, on March 30, 2017. Our target number of animals was 1200 (30 strains X 2 sexex X 4 treatments X n = 5); however, 1050 samples proved to have RNA suitable for analysis.

### 2.2. Materials

The following drugs and chemicals were obtained from the sources indicated: DFP, (Sigma, St. Louis, MO, USA), CORT (Steraloids, Inc., Newport, RI, USA). All reagents were analytical grade.

## 3. Experimental Design

### 3.1. Treatment Groups

The animals were divided into four treatment groups as follows:Control. Plain tap water for fluid, saline injection and euthanized 6h after injection by cervical dislocation followed by decapitation. The brain was removed, and the frontal cortex dissected, weighed and placed on dry ice and stored at −80 °C until assay for cytokine gene expression.CORT group. These animals received tap water containing 20 mg% CORT dissolved in 0.6% (*v/v*) EtOH vehicle for 8 days. On the 8th day, the animals were injected with saline, euthanized 6h after injection by cervical dislocation followed by decapitation. The brain was removed, and the frontal cortex dissected, weighed and placed on dry ice and stored at −80 °C until assay for cytokine gene expression.DFP group. These animals received plain tap water for fluid and were injected with 4 mg/kg DFP, i.p. 6 h after injection, the animals were euthanized by cervical dislocation followed by decapitation. The brain was removed, and the frontal cortex dissected, weighed and placed on dry ice and stored at −80 °C until assay for cytokine gene expression.CORT-DFP (C + D), group. These animals received tap water containing 20 mg% CORT dissolved in 0.6% (*v/v*) EtOH vehicle for 8 days. On the 8th day, the animals were injected with 4 mg/kg DFP, i.p. 6 h after injection, and the animals were euthanized by cervical dislocation followed by decapitation. The brain was removed, and the frontal cortex dissected, weighed and placed on dry ice and stored at -80 °C until assay for cytokine gene expression.

An abbreviated version of the design is presented in Table 1.

### 3.2. RNA Isolation, cDNA Synthesis and rtPCR

QPCR was used to analyze expression of mRNA for the proinflammatory cytokines, Il1β, Il6, and *Tnfa* in brain samples (medial prefrontal cortex). All procedures are described by O’Callaghan et al. [10] and Locker et al. [8]. Total RNA was isolated from medial prefrontal cortex at 6 h after DFP exposure. Real-time PCR analysis of the housekeeping gene, glyceraldehyde-3-phosphate dehydrogenase (GAPDH), and of the proinflammatory mediators, TNFα, IL-6, and IL-1β was performed in an ABI7500 Real-Time PCR System (Thermo Fisher Scientific, Waltham, MA, USA) in combination with TaqMan chemistry. Relative quantification of gene expression was performed using the comparative threshold (ΔΔC_T_) method. Changes in mRNA expression levels were calculated after normalization to GAPDH. The ratios obtained after normalization are expressed as fold change over corresponding saline-treated controls.

### 3.3. Data Analysis

Transcript abundance for *Il1β, Il6,* and *Tnfa* obtained by qPCR were LOG_2_ transformed and analyzed by analysis of variance for a three between-subjects variables (strain, sex, treatment) experiment. Main effects and interactions were considered statistically significant at α = 0.05. Genetic mapping and quantitative trait loci (QTL) analyses were conducted using GeneNetwork software (http://www.genenetwork.org; [16]). The data of greatest importance to this study are from the CORT-DFP treatment group; however, we also present the findings from the CORT alone and DFP alone treatment groups.

## 4. Results

### 4.1. Corticosterone Consumption

Average consumption of CORT added to the drinking water over the seven days varied widely (Figure 1). Analysis of variance revealed strain and sex main effects (F_33,524_ = 12.12, *p* < 0.001; F_1,524_ = 91.45, *p* < 0.001, respectively). Overall, females (GeneNetwork ID 21265) consumed more corticosterone than did the males GeneNetwork ID 21273). The interaction between strain and sex was also significant (F_33,524_ = 1.77, *p* < 0.01). We also evaluated the effect of variability in corticosterone consumption on expression of the three cytokines by conducting analysis of covariance and reporting the adjusted means.

### 4.2. Gene Expression in Response to Treatments

#### 4.2.1. IL1b

We observed wide variability in the effect of CORT on expression of *Il1b* (Figure 2). ANOVA revealed a significant main effect for strain (F_32,122_ = 3.61, *p* < 0.001). The main effects for sex and the sex X strain interaction were not significant (F < 1 for both). Interestingly, CORT increased the expression of this cytokine in BXD strains 78 and 79 in males and 60 and 89 in both sexes.

DFP produced inconsistent effects on *Il1b* expression (Figure 3); however, the combination of DFP with CORT increased the expression in nearly all strains, thus supporting the observation of O’Callaghan and colleagues [10] that CORT enhances the expression of this cytokine and others in the prefrontal cortex. Analysis of variance revealed significant effects for strain, sex, and treatment (F_33,843_ = 2.56, *p* < 0.001; F_1,843_ = 4.67, *p* < 0.04; F_3,843_ = 253.31, *p* < 0.001 respectively) on *IL1b* expression. The strain X treatment, sex X treatment and sex X strain interactions were also significant (F_97,843_ = 1.40, *p* < 0.02: F_3,843_ = 3.76, *p* < 0.02; F_32,843_ = 2.51, *p* < 0.001, respectively). Analysis of covariance showed no significant effect of CORT consumption (F_1,250_ < 1).

Interestingly, CORT-DFP was the only condition that showed no decreases in expression (Figure 3) and in the same treatment condition; females (GeneNetwork ID 21195) were less affected in increased gene expression than males (GeneNetwork ID 21200).

#### 4.2.2. IL6

We observed strain and sex-related variability in the effects of CORT on expression of *Il6* (Figure 4). It appears that males in only one BXD strain (BXD75) showed an increase in expression of this cytokine.

Analysis of variance revealed significant main effects for strain and sex (F_32,112_ = 1.92, *p* < 0.01; F_1,112_ = 5.92, *p* < 0.02), The strain X sex interaction was not significant (F < 1). Overall, females showed a greater effect of CORT than did males.

We observed inconsistent effects of DFP and DFP+CORT on expression of *Il6* (Figure 5). Effects for females are seen on the top panel and data for males on the bottom panel. Analysis of variance revealed significant effects for strain, sex, and treatment (F_33,843_ = 2.51, *p* < 0.001; F_1.843_ = 7.91, *p* < 0.006; F_3,843_ = 16.01, *p* < 0.001 respectively). None of the 2-way interactions was significant (F < 1 for all) and the three-way interaction was also not significant (F < 1). As with *Il1b*, females were less sensitive to the CORT + DFP effect on Il6 gene expression. Analysis of covariance showed no significant effect of CORT on *Il6* gene expression (F_1,249_ = 2.12, *p* > 0.1).

#### 4.2.3. Tnfa

CORT produced variable effects on *Tnfa* expression (Figure 6) among the strains, compared to controls. Strain means can be found in GeneNetwork for males (GeneNetwork ID 21234) and females (GeneNetwork ID 21228) respectively. The data are Log_2_ means for gene expression by qPCR.

ANOVA revealed a significant main effect for strain (F_32,111_ = 5.31, *p* < 0.01). Effects for sex or strain X sex interaction were not significant (F_1,111_ = 1.83, *p* < 0.20; F_30,111_ = 1.98, *p* < 0.35).

DFP and DFP + CORT also affected *Tnfa* expression in prefrontal cortex (Figure 7). Analysis of variance revealed significant effects for strain, sex, and treatment (F_33,843_ = 3.51, *p* < 0.001; F_1.843_ = 17.26, *p* < 0.001; F_3,843_ = 269. 75, *p* < 0.001 respectively). The strain X treatment, sex X treatment and sex X strain interactions were also significant (F_97,843_ = 1.52, *p* < 0.002; F_3,843_ = 3.30, *p* < 0.02; F_32,843_ = 2.19, *p* < 0.001, respectively). Analysis of covariance showed a significant effect of CORT consumption on CORT+DFP gene expression (F_1,249_ = 5.10, *p* < 0.03). As with the other two cytokines, females were less sensitive to the combined effects of CORT+DFP. Accordingly, Figure 7 presents adjusted means for expression following CORT + DFP treatment. The data for females are presented in the top panel and the data for males in the bottom.

### 4.3. Mapping of IL1β Response to CORT + DFP

Of the three cytokines, *Tnfa* and *Il1β* showed significant strain (genetic) X treatment interactions. Mapping *Il1β* response, we observed a suggestive quantitative trait loci (QTL) on chromosome 7 between 110 and 115 Mb for males and females. When we combined the male and female data by principal component analysis (PCA), the signal became significant (Figure 8). We then searched for possible candidate genes that underlie the individual differences in *Il1β* response to CORT+DFP using (*Hippocampus Consortium M430v2 (Jun06) PDNN Genenetwork accession no. GN112*). In this database, the overlap of strains with our study included 21 strains. The main criteria are 1) that the gene be *cis-*regulated (not absolutely necessary) and 2) the expression of the gene be correlated with variability in the phenotype (principal component Figure 9). One gene stood out, *Spon1* (Spondin 1). The protein is secreted by the floor plate during neurogenesis and is involved in cell adhesion, axon guidance, metal binding (Ca), LPS-related inflammation, amyloid precursor protein degradation, negative regulation of amyloid beta production.

In investigating the pattern of expression across the gene between the parental B6 and D2 strains, we noticed that the 3′ UTR region of the gene (Figure 10) showed greater expression in the B6 strain (in blue extreme right side of figure) compared to the D2 strain. This part of the gene is important because it usually contains regulatory sequences that affect expression.

## 5. Discussion

Our work presented here is the first of its kind to show genetic-based differential susceptibility in a mouse model of GWI that may underlie the symptoms observed in ill Gulf War veterans. This work also shows sex differences with females being less sensitive to the effects of the exposure than males. Moreover, our results demonstrate once again the efficacy of our model in terms of CORT enhancing the effect of DFP on expression of proinflammatory cytokine genes. Please note that this work addresses the acute response of proinflammatory cytokines to CORT + DFP as proof-of-principle. GWI is a chronic disease and those so afflicted have suffered symptoms for nearly 30 years. Study of the chronic effects, susceptibility and treatments is compelling and likely related to epigenetics of genes identified here. Indeed, Trevedi and colleagues [17] showed DNA methylation changes in monocytes from GWI affected veterans in genes related to immune function, thus supporting the hypothesis of neuroimmune sequelae of GWI. In mice, Ashbrook et al. [18] showed genome-wide histone modification and DNA methylation in the frontal cortex. Many of the genes so modified are related to myelin production in oligodendrocytes and thus may be related to the observation of reduced white matter in frontal cortex of GWI veterans.

The importance of this work is that candidate genes identified in the mouse have high probability of overlapping with the human genome [19,20] because of conserved function (gene homology) and biological pathways.

In comparing the phenotypic responses, *Il1β* and *Tnfa* were better indices of the treatment effect of DFP following corticosterone than was *Il6*, in agreement with our prior findings [10]. Both of the former showed the expected main effects, but more importantly the interactions between strain and treatment, and sex by treatment. None of the 2-way interactions for Il6 were significant.

We were also able to nominate a candidate gene for *Il1β* response to treatment. *Spon1* met both criteria of *cis-*regulation and tight correlation with the response. *Spon1* is of interest because it is involved with axon guidance, WNT/βcatenin signaling and protects against chemotherapy toxins [20,21], cognitive problems [22], circadian rhythms [23] and TGF-β- inflammatory response [24]. We observed differential expression of the distal 3′ UTR between C57BL/6J and DBA/2J (Figure 10). This region contains binding sites for several RNA-binding proteins. Higher expression of the B allele relative to the D allele may lead to differential transport of this mRNA in neurons across BXD strains. The B2 SINE polymorphism in the 3′ UTR of Comt is an example the key role of these types of non-coding splice variants [25].

There are other things that draw our attention to *Spon1. One* is the observation by Zhao and colleagues [21] that R-Spondin 1 reduces the toxicity of chemotherapeutic agents and radiation via the WNT/βcatenin pathway [26]. The mechanism may involve changes in proinflammatory cytokine production locally and it is worth exploring whether Spon 1 might affect other inflammatory systems, including IL1 cytokines. If so, then Spon 1 and WNT/βcatenin might be appropriate targets for therapeutics for GWI. Another and more compelling reason is its effects on amyloid precursor protein. Recent genome-wide association studies of the rate of cognitive decline in Alzheimer’s disease indicated SPON1 as candidate gene associated with slower rate of cognitive decline in Alzheimer’s disease [27]. Hafez and colleagues [28] proposed from their study in mice that one possible mechanism is *Spon1* associated with inhibition of amyloid beta while promoting synaptophysin. 

## 6. Conclusions

We have demonstrated wide, genetic and sex variability in susceptibility to developing GWI following combined exposure to high circulating glucocorticoids and organophosphorus compounds that inhibit cholinesterase irreversibly. By locating a plausible candidate gene that underlies the strain differences in a genetic reference population of mice, we may have identified a possible factor underlying individual differences to the conditions that produce GWI, especially as concerns cognitive difficulties in a genetically defined subpopulation of humans. The study has its limitations. First, we were constrained by budgetary considerations to studying just three proinflammatory cytokine genes. A broader panel of these genes would give a more complete view. Second, we were again limited to studying 30 BXD strains. By studying more strains, we would expect to discover more candidates and with increased precision in mapping. We did perform whole genome RNA-seq analysis and those data will be submitted for publication separately.

## Figures and Tables

**Figure 1 brainsci-10-00143-f001:**
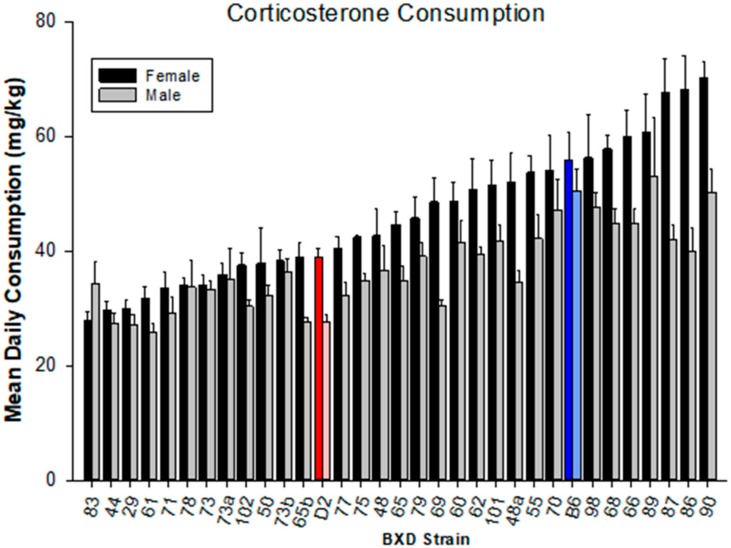
Corticosterone consumption. Male and female BXD mice were given 20mg% (*w/v*) corticosterone in their drinking water as sole liquid source for seven days prior to i.p. treatment with 4 mg/kg diisopropylflurophosphate. Data are mean consumption per day ± s.e.m.

**Figure 2 brainsci-10-00143-f002:**
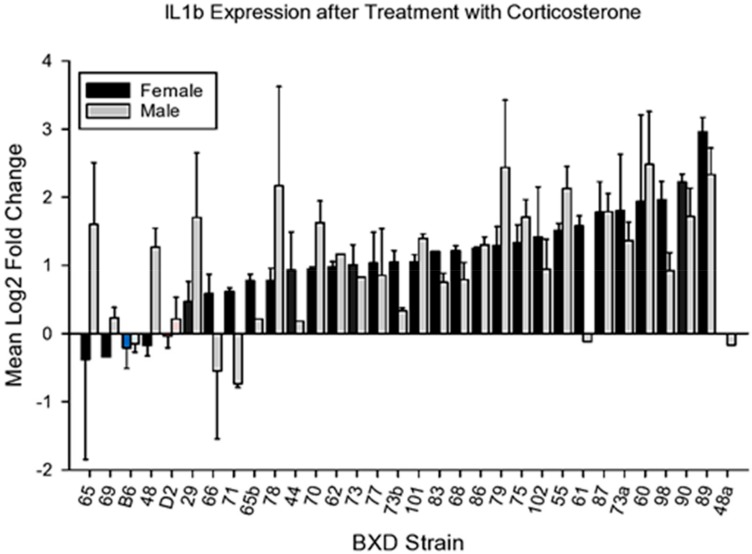
Mean change (±s.e.m) vs. control in *Il1b* expression in prefrontal cortex following 7 days of corticosterone in the drinking water and 6h following intraperitoneal injection with normal saline.

**Figure 3 brainsci-10-00143-f003:**
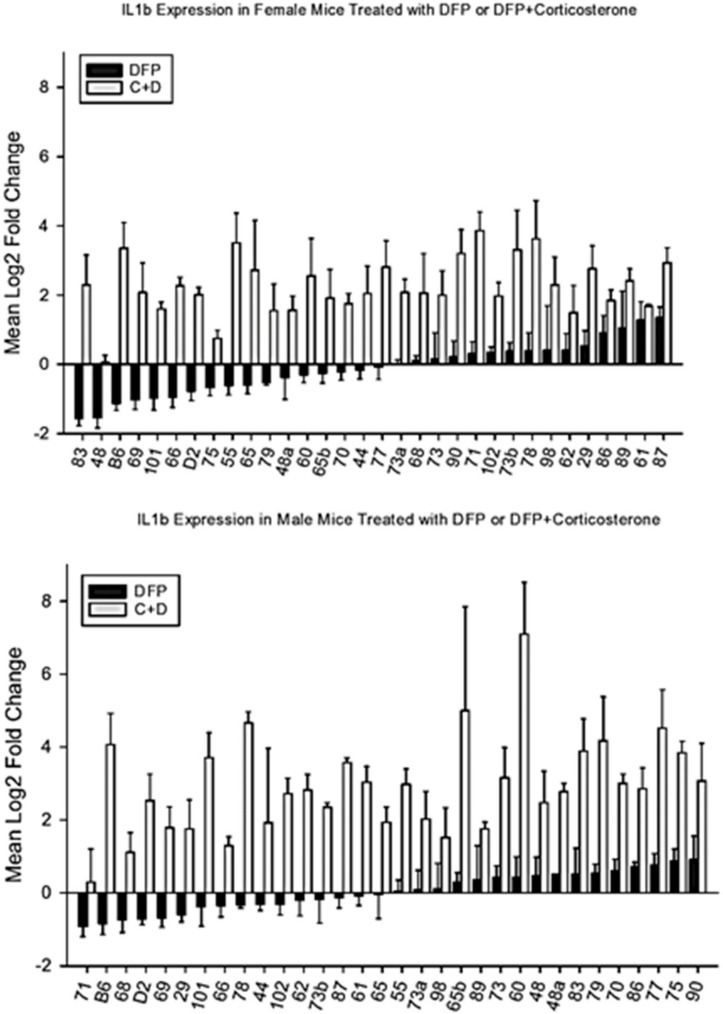
Mean change (±s.e.m) vs. control in *Il1b* expression in prefrontal cortex 6 h following intraperitoneal injection with diisopropylflurophosphate (DFP, 4 mg/kg—black bars) or 7 days of corticosterone in the drinking water and 6h following intraperitoneal injection with diisopropylflurophosphate (C + D, gray bars).

**Figure 4 brainsci-10-00143-f004:**
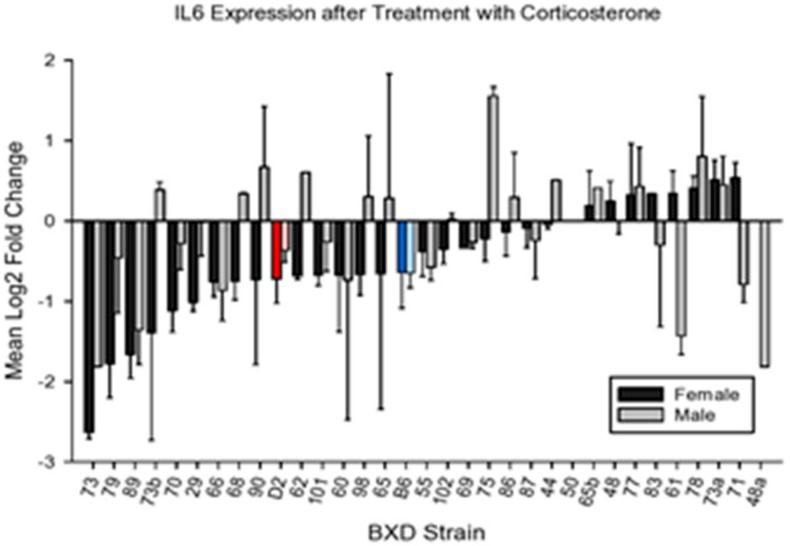
Mean change (±s.e.m) vs. control in *ll6* expression in prefrontal cortex following 7 days of corticosterone in the drinking water and 6h following intraperitoneal injection with normal saline.

**Figure 5 brainsci-10-00143-f005:**
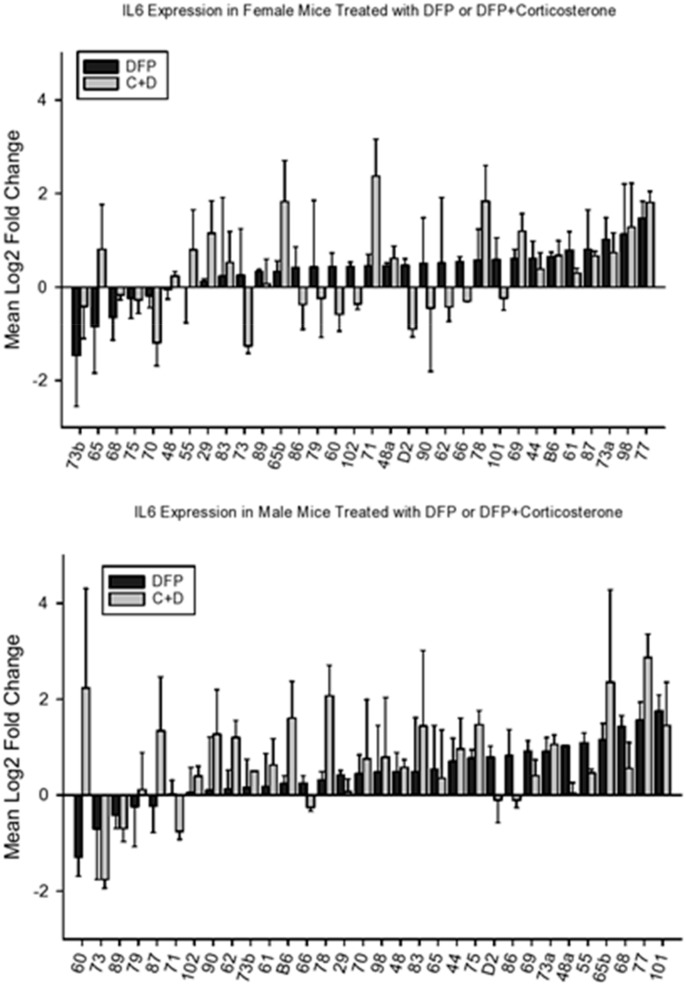
Mean change (±s.e.m) vs. control in *Il6* expression in prefrontal cortex 6 h following intraperitoneal injection with diisopropylflurophosphate (4 mg/kg—black bars) or 7 days of corticosterone in the drinking water and 6h following intraperiteoneal injection with diisopropylflurophosphate (gray bars). Top panel, females, bottom panel, males.

**Figure 6 brainsci-10-00143-f006:**
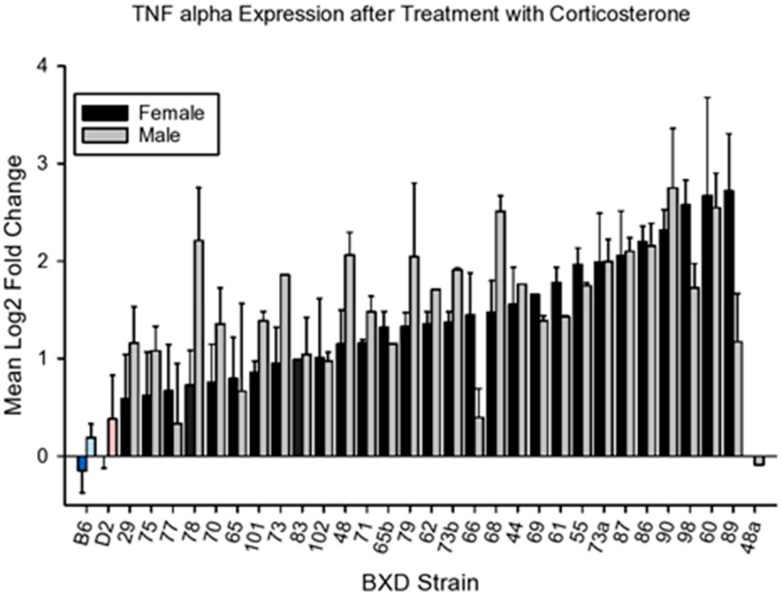
Mean change (±s.e.m) vs. control in *Tnfa* expression in prefrontal cortex following 7 days of corticosterone in the drinking water and 6h following intraperitoneal injection with normal saline.

**Figure 7 brainsci-10-00143-f007:**
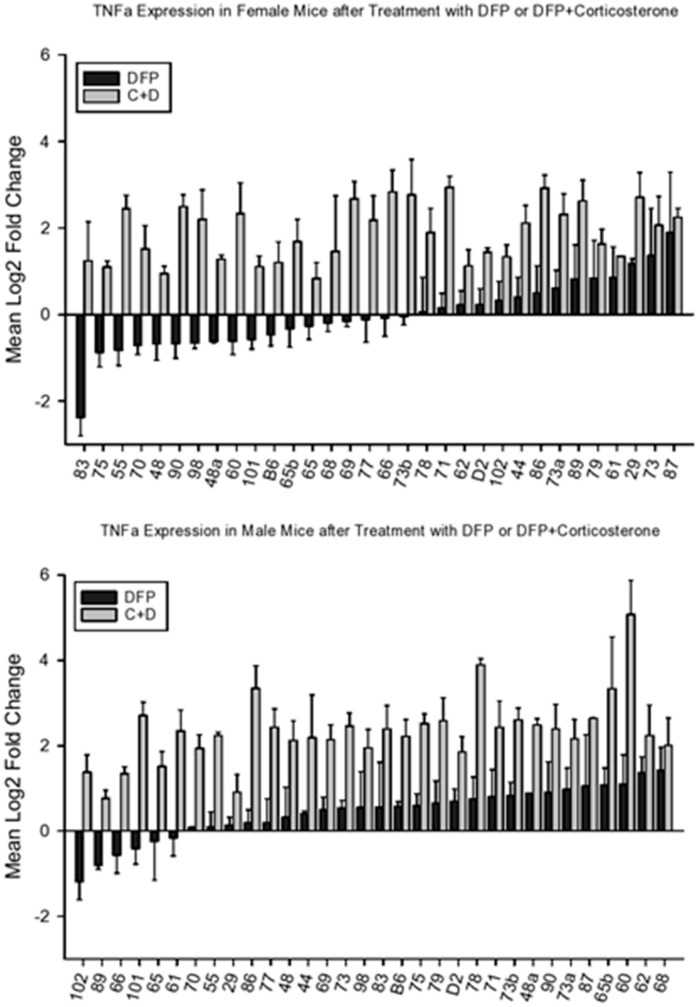
Mean change (±s.e.m) vs. control in *Tnfa* expression in prefrontal cortex 6 h following intraperitoneal injection with diisopropylflurophosphate (4 mg/kg—black bars) or 7 days of corticosterone in the drinking water and 6 h following intraperiteoneal injection with diisopropylflurophosphate (gray bars). Top panel, females, bottom panel, males.

**Figure 8 brainsci-10-00143-f008:**
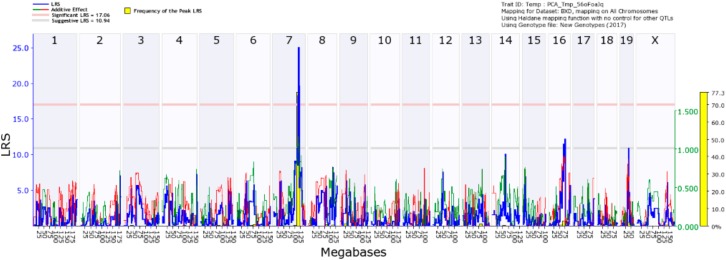
Quantitative trait locus map of *Il1b* expression change following 7 days of corticosterone in the drinking water and 6h following i.p. injection of diisopropylflurophosphate (4 mg/kg). The dependent variable (eigenvariable) is the first principal component derived from males and females individually and combined. Genome-wide interval mapping was performed using GeneNetwork software. The map shows a significant peak on chromosome 7 and two suggestive peaks, one on chromosome 16 and the other on chromosome 19. The latter two peaks lacked sufficient bootstrap support (yellow bars) and thus considered spurious.

**Figure 9 brainsci-10-00143-f009:**
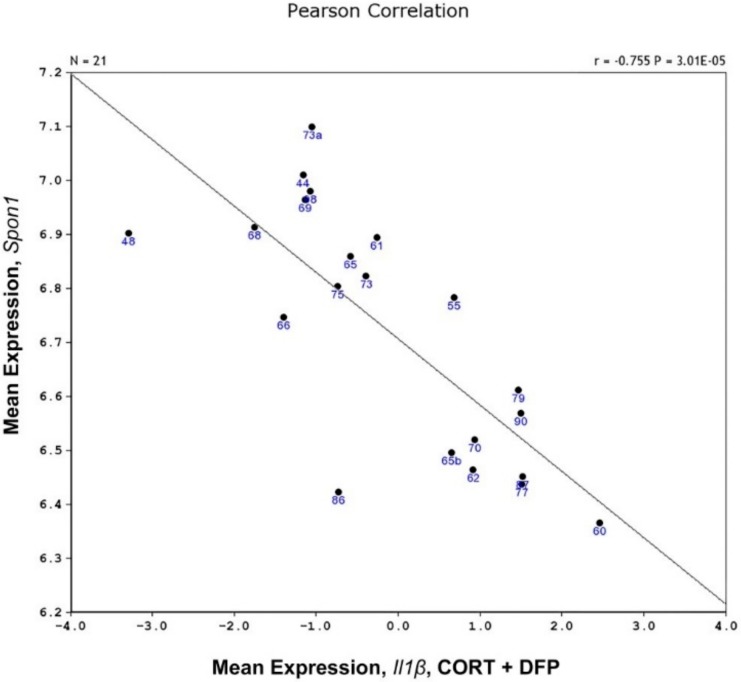
Scatter diagram of the association between the expression of *Spon1* and the eigenvariable relative to *Il1b* expression in mice treated with corticosterone and DFP.

**Figure 10 brainsci-10-00143-f010:**
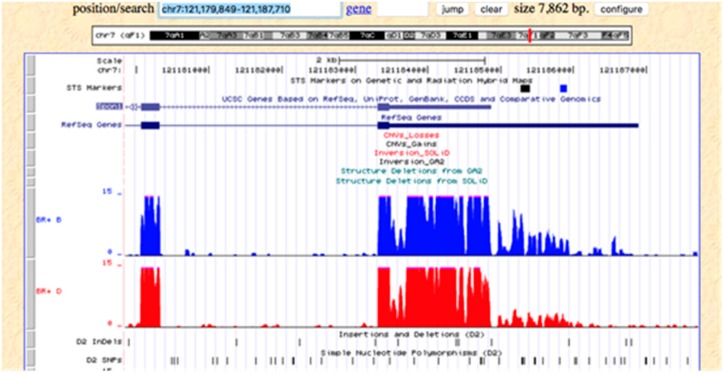
Diagram of *Spon1* expression from 5′ to 3′ UTR (left to right) the top (blue) figure is C57BL/6J and the bottom (red) figure is DBA/2J. The figure shows greater expression in the 3′ UTR for C57BL/6J compared to that for DBA/2J.

**Table 1 brainsci-10-00143-t001:** Experimental design of the study by treatment groups.

**Control**	Day 1–7 plain water; day 8 saline injection followed 6 h later by PFC harvest
**CORT**	Day 1–7 Corticosterone in drinking water; day 8 saline injection followed 6h later by PFC harvest
**DFP:**	Day 1–7 plain water; day 8 DFP injection followed 6 h later by PFC harvest
**C + D**	Day1–7 Corticosterone in drinking water; day 8 DFP injection followed 6h later by PFC harvest

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
