# Peer review of "Modeling the Genetic Basis of Individual Differences in Susceptibility to Gulf War Illness"

_brainsci, 2020, doi:10.3390/brainsci10030143_

Round 1

Reviewer 1 Report

Authors are exploring a genetic basis for individual differences in susceptibility to Gulf War Illness disease in a series of 30 inbred mouse strains that were exposed to OP compound DFP along with a glucocorticoid corticosterone. Authors report gender and strain differences and finally identify Spon1 as the candidate gene for the observed differences to DFP and CORT exposures.

Following questions must be addressed:

The reported changes could indicate acute effects of CORT+DFP exposures but may not be representative of its chronic effects since the mice are sacrificed 6-h after exposure. Explanation is needed why this time point was selected and how it is representative of GWI phenotype.

A general question is whether or not a hierarchical (mixed effects) model was used to account for the fact that individual mice are nested within strains in these analyses? Is that automatically accounted for in the GeneNetwork software? More specifically, the lack of methodological detail for the"Mapping of IL1β response to CORT+DFP" section on p10 and corresponding Fig.8 is a concern.

While the use of BXD lines to map genomic regions for specific phenotypes is a completely standard and reasonable approach, the response phenotype being analyzed is somewhat opaque. How was this analysis set up? Was response entered per mouse or per BXD line? For example, was response analyzed as raw IL1β expression levels per mouse with treatment group coded as a dummy variable or was response analyzed as an average fold change by strain? 

Furthermore, it is unclear how the males and females could be "combined by principal components analysis (PCA)" p10 to achieve a significant result. The lack of detail makes it very hard to judge if the analysis was appropriate.

Author Response

Responses to Reviewer 1: 

 Concern about the acute nature of the response. The reported changes do indicate acute effects of all treatments, including CORT+DFP. This timepoint was selected as proof-of-principle that the combined exposures produce neuroinflammation and the basis for sickness behavior (see the Dantzer reference). We have work in progress to show that the exposures produce long term differential, genome-wide methylation to address the chronic nature of the illness. The model used (O’C 2015) was developed to model in theater and emerging theater exposures.  GWI emerged immediately post the 1991 GW and persists, thus, the present design was to begin to characterize the genetic susceptibility of veterans as the return from deployment with GWI  Concern about the design. The design was a simple, completely crossed factorial design with three between-subjects variables, viz., strain, sex and treatment. The ANOVA was performed using SPSS software.  Mapping of IL1b.  First, we crated an eigenvariable for sexes separately and combined and then performed interval mapping using the GeneNetwork software for the eigenvariable means. The procedure involves selecting the GN phenotypes of interest, produce a correlation matrix and then selecting principal component analysis. A temporary eigenvariable is produced with strain distribution of the means and then selection of mapping of those means. The point is to take several related phenotypes and reduce them to one index of response to treatment (eigenvariable).  The result is nearly always an increase in the signal for the combined variable compared to the individual phenotypes.  Means vs. individual response.  As indicated in the graphs (ordinates), legends and text, all of our variables were expressed as means and s.e.m (for graphs).  Unclear as to how male and female data could be combined.  One of the really useful features of GN is the feature to create correlation matrices for multiple phenotypes. One selects the phenotype of interest, then select find correlated measures (phenotypes, gene expression, etc. select those for correlation matrix and then select perform principal components analysis. The result is a strain distribution listing that can be treated as any other phenotype, including mapping. 

Reviewer 2 Report

Overview:  The data presented in this paper represent a tiny subset from a much larger and interesting study of gene expression across multiple different inbred strains of mice in response to exposure to a paradigm that has been used as a model of Gulf War Illness, namely treatment with corticosterone and the cholinesterase inhibitor DFP.  The results of that study may yield interesting candidate genes which confer influence on outcomes following exposure to DFP.  From the point of view of GWI, given the heterogeneity of exposures suffered by our troops, a more translationally relevant study might have been to investigate multiple different exposure paradigms, in fewer strains, and then expand the study to more strains once common targets were identified.  Nonetheless, from that large study, the data from which are described as forthcoming, the work described herein is simply an analysis of expression of IL1b, TNFa and IL6 in the pre-frontal cortex of the mice from that study.   This has the potential to present a proof of concept by identifying genetic influences on response to GW agent exposure, but as currently analyzed the data are insufficient.  Specifically it is unclear why, having generated the expression data for IL1b, TNFa and IL6, the authors did not appropriately present the data to enable direct comparison of the most relevant groups (I.e. DFP+Cort versus Cort alone), nor why the results from TNFa (which, like IL1b showed strain x treatment interaction) were not mapped, nor why only one of three peaks for IL1b was subject to further investigation to identify a gene of interest.  Even if the data for IL6 did not show an interactive effect, the data from IL1b had the potential to identify at least two other candidate genes, and from TNFa again candidate genes could have been identified for any relevant chromosome locations.  Therefore the focus on spondin is inappropriate.

Specific comments:

The introduction is primarily an introduction to the O'Callaghan model rather than to the potential role of any genetic influences on GWI presentation and severity.

The Materials and Methods are insufficiently described, with not even a general explanation given but only, at best, references to previous work.  With a documented 5-8 mice per strain, sex and treatment group, 30 strains and 4 treatments, the main study appears to have required 1200-1920 mice - is this correct? This should be stated.

In the Results the graphs should be harmonized, with the strains presented in the same order, explanation of the color coding (D2 and B6?), and comparison of the most RELEVANT groupings - for this study the main response to exposure is in the comparison of DFP+CORT versus CORT alone - not DFP+CORT versus DFP as presented.  It appears that results were standardized for cort consumption but this could be made more clear as it is an important consideration.  It would be possible, without graphs becoming overly complicated, to plot all three treatment groups together, separating by sex.  Why was the TNFa response not mapped?  Why were the other two chromosomal locations (c16 and c19) not investigated for the IL1b response? How was cis regulation determined?  The Figure 9 Y axis label is incomprehensible and only 21 strains appear to be plotted - what has happened to the other 9 strains?  Figure 10 is a screenshot that looks untidy and does not add to the value of the paper - the data supposedly represented therein are not terribly clear and could be described in text.

In the Discussion, even though only one out of a minimum of 3 genes has been presented, much has been left unaddressed.  How does spondin change across strains?  across species?  Are there polymorphisms in the human spondin gene that could be investigated in candidate gene studies in human GWI patient populations versus controls?

Author Response

Responses to Reviewer 2:

Concern that the data analysis is insufficient. In fact, we are following the model developed by O’Callaghan that shows CORT+DFP to be the most translationally relevant treatment, compared to control to model exposures that occurred as veterans returned from theater with GWI Il1b gave us the only significant peak. The highest LRS for TNFa was 11 and not suggestive. Focus on Spondin 1 is inappropriate. We respectfully disagree. We draw your attention to lines 246-265 and moreover, our recent findings with RNA-seq confirm that Spondin 1 is indeed responsive to CORT+DFP.  These data are in preparation for submission in a separate manuscript. The introduction is primarily an introduction to the O’Callaghan model rather than to the potential role of any genetic influences. Response: Lines 1-46 out of 67 in the introduction describe GWI Lines 56-67 describe the utility of the BXD RI family as a model to track down the genetics of susceptibility Lines 46-56 describe the O’Callaghan model and why it is important. The most relevant comparison is DFP+CORT vs. DFP (according to the literature) not DFP+CORT vs CORT. As explained in the manuscript, CORT consumption as a covariate related to TNFa only. Figure 9 Y axis label is incomprehensible. This is the name of the gene expression file for Spon1 in hippocampus as described in the text, line 209.  We have changed the labels to be more comprehensible. There are only 21 data points in Figure 9. What happened to the other 9? The gene expression data were generated by the Hippocampus Consortium in 2006 (full description is available on www.GeneNetwork.org) and only 21 strains from this study overlapped with ours. Our target number of animals was 1200 but we obtained adequate RNA from 1050 mice. Why was the TNFa response not mapped? Because the highest LRS for this phenotype was 11 and far from even suggestive. Why were the other two peaks (Chr 16 and 19) not investigated? Look at Figure 8.  There is practically no bootstrap support (yellow bars) for either peak, thereby rendering them spurious. Are there polymorphisms in human spondin? Sure enough. Check out lines 255-265. A GWAS study by Sherva and colleagues (2014 and cited) shows Spon1 to be associated with rate of cognitive decline in humans.  Other species? Terra incognita.

Round 2

Reviewer 1 Report

Authors have satisfactorily answered my queries.

However, including these responses in the manuscript will improve the readability of the paper. For example: including their description for the acute versus chronic nature of CORT+DFP exposures in the discussion will help the readers further understand this proof-of-concept study and appreciate future scope of the study. Similarly, expanding on the application of the genenetwork tool will be helpful.

Author Response

We have addressed Reviewer 2's request to address the problem of acute vs. chronic effects of CORT+DFP exposure in the Discussioon (red typeface) and elucidation of how to use Genenetwork in the Methods (red typeface).